# Experimental Study on Compatibility of Human Bronchial Epithelial Cells in Collagen–Alginate Bioink for 3D Printing

**DOI:** 10.3390/bioengineering11090862

**Published:** 2024-08-23

**Authors:** Taieba Tuba Rahman, Nathan Wood, Yeasir Mohammad Akib, Hongmin Qin, Zhijian Pei

**Affiliations:** 1Department of Industrial & Systems Engineering, Texas A&M University, College Station, TX 77843, USA; yeasir.akib@tamu.edu (Y.M.A.); zjpei@tamu.edu (Z.P.); 2Department of Biology, Texas A&M University, College Station, TX 77843, USA; woodn@tamu.edu (N.W.); hqin@tamu.edu (H.Q.)

**Keywords:** collagen–alginate bioink, compatibility, cell viability, culture well method, human bronchial epithelial cells

## Abstract

This paper reports an experimental study on the compatibility of human bronchial epithelial (HBE) cells in a collagen–alginate bioink. The compatibility was assessed using the culture well method with three bioink compositions prepared from a 10% alginate solution and neutralized TeloCol-10 mg/mL collagen stock solution. Cell viability, quantified by (live cell count—dead cell count)/live cell count within the HBE cell-laden hydrogel, was evaluated using the live/dead assay method from Day 0 to Day 6. Experimental results demonstrated that the collagen–alginate 4:1 bioink composition exhibited the highest cell viability on Day 6 (85%), outperforming the collagen–alginate 1:4 bioink composition and the alginate bioink composition, which showed cell viability of 75% and 45%, respectively. Additionally, the live cell count was highest for the collagen–alginate 4:1 bioink composition on Day 0, a trend that persisted through Days 1 to 6, underscoring its superior performance in maintaining cell viability and promoting cell proliferation. These findings show that the compatibility of HBE cells with the collagen–alginate 4:1 bioink composition was higher compared with the other two bioink compositions.

## 1. Introduction

Three-dimensional bioprinting involves the precise layer-by-layer deposition of biocompatible materials and cells to fabricate functional biological constructs in a predesigned manner for tissue engineering or other biological studies [1]. In respiratory research, 3D bioprinting has shown immense potential, particularly in developing lung tissue models for in vitro drug testing and therapeutic interventions for respiratory diseases. For instance, da Rosa et al. demonstrated the fabrication of a bioprinted 3D construct with lung epithelial cells (alveolar type I and II, ciliated, and secretory cells) for in vitro drug testing against COVID-19 and other respiratory diseases [2]. Wang et al. bioprinted a 3D construct with lung cancer cells A549 and 95-D for lung cancer research [3]. These studies underscore the potential of 3D bioprinting in creating functional lung models, enhancing the study and treatment of respiratory diseases [4].

Bioinks used in bioprinting provide the necessary environment for cell growth and tissue formation [5]. Ensuring the compatibility of bioinks is crucial to avoid adverse cellular responses and to promote optimal cell viability, proliferation, and differentiation [6]. Cell viability is defined as the ability of cells to survive and maintain metabolic activity under given conditions [7]. Cell proliferation is defined as the increase in cell number resulting from cell growth and division [8]. The study of bioink compatibility is vital in identifying materials that can effectively support the specific requirements of various cell types.

Table 1 shows the bioinks used in reported studies on bioprinting with lung cells, with alginate being one of the main bioinks used. Alginate (also known as sodium alginate), a naturally occurring polysaccharide derived from brown seaweed, is commonly used in bioink formulations due to its ability to form hydrogels in the presence of divalent cations, such as calcium ions [5]. Moreover, its biocompatibility, non-toxic nature, and low cost make it an attractive material for bioprinting applications [9]. However, alginate hydrogels are inherently bioinert, lacking the cell adhesion sites necessary for cellular attachment and proliferation [5,9,10]. To address this limitation, alginate is often combined with collagen, the primary structural protein in the extracellular matrix (ECM). Collagen provides the necessary biochemical cues to promote cell adhesion, proliferation, and differentiation [11,12]. Among the 29 types of collagen available, types I–III constitute 80–90% of total body collagen [13]. Biomaterials in tissue engineering studies mostly use type I collagen, which is the main component in the connective tissues of mammals [11,14,15,16].

Collagen–alginate hydrogel was selected for this study because it has some different features in comparison with polyethylene glycol (PEG) diacrylate, agarose, gelatin, and hyaluronic acid. Collagen, as a natural extracellular matrix component, enhances cell attachment and growth, while alginate contributes to improved printability and structural integrity through rapid gelation with an ionic crosslinking solution. In contrast, the PEG diacrylate bioink requires photo crosslinking to form a hydrogel [23]. Photo crosslinking involves photoinitiator reagents and UV light that might damage the cells, potentially reducing cell viability [24]. Although agarose offers biocompatibility and strong mechanical properties, it is bioinert and lacks cell-adhesive properties [25,26]. Gelatin, despite supporting cell adhesion, requires precise temperature control due to its thermo-gelling properties [23,27,28]. Moreover, it forms hydrogel at low temperatures (e.g., 10–30 °C), which may damage the cells [26]. Hyaluronic acid, although biocompatible and hydrophilic, lacks mechanical strength, leading to poor structural integrity [29].

Reported studies have demonstrated the efficacy of the collagen–alginate bioink in enhancing the mechanical and biological properties required for tissue engineering applications. For instance, Zimmerling et al. synthesized the bioink from alginate and collagen for bioprinting respiratory tissue models using human pulmonary lung fibroblast (HPF) cells [12]. Perez et al. fabricated a dual-layered fibrous structure as a cell delivery system for bone tissue engineering, composed of collagen and alginate as the core and shell, respectively, using bone marrow mesenchymal stem cells [30]. These investigations show the combined benefits of alginate’s mechanical stability and collagen’s bioactivity, making the alginate–collagen bioink a promising choice for various tissue engineering endeavors. Despite these advancements, there remains a notable gap concerning the compatibility of this bioink with specific cell types, particularly human bronchial epithelial (HBE) cells.

Human bronchial epithelial (HBE) cells are critical in forming the lining of the respiratory tract. They are involved in various physiological functions, including mucociliary clearance and barrier formation. These functions are vital for maintaining respiratory health by removing mucus and pathogens from the respiratory tract and protecting underlying tissues from harmful substances [31]. However, to date, there has been a lack of comprehensive studies evaluating the compatibility of HBE cells in collagen–alginate bioink. This study aims to fill this gap by investigating the compatibility of HBE cells in collagen–alginate bioink.

## 2. Materials and Methods

### 2.1. Cell Culture

16HBE14o-human bronchial epithelial cells (Cat. No. SCC150), isolated from a 1-year-old male heart–lung patient, were purchased from Millipore Sigma (Sigma Cat. No. M2279, Saint Louis, MO, USA). Cryopreserved cells were cultured according to the supplier’s instructions. The cell culturing process is illustrated in Figure 1. At first, the vial containing frozen 16HBE14o cells was removed from liquid nitrogen storage and put in a water bath at 37 °C. As soon as the cells were completely thawed, the outside of the vial was disinfected with 70% ethanol, and the vial was placed in a biosafety cabinet (SterilGARD II Advance, Baker, Sanford, ME, USA). The cells were transferred using a 1 mL pipette to a sterile 15 mL conical tube. Using a 10 mL pipette, 9 mL of α-MEM medium (cell growth medium) was slowly added dropwise into the conical tube with the cells. The cell suspension with the medium in the conical tube was mixed gently using a pipette to draw the cell suspension with the medium into the pipette and then expel it back into the conical tube, repeating this process twice to ensure thorough mixing without damaging the cells. The conical tube was then centrifuged at 900 rpm for 2–3 min to pellet the cells and separate the supernatant. Then, the supernatant was gently decanted as much as possible from the conical tube. These steps were necessary to remove residual cryopreservatives. Then, 10 mL of α-MEM medium was resuspended slowly into the conical tube with the cells. The cell mixture was transferred to an ECM-coated T75 tissue culture flask. Then, the flask with the cells was placed in a humidified incubator at 37 °C with 5% CO_2_. The next day, the medium in the flask was removed from the flask, and 10 mL of fresh α-MEM medium was added in. Every other day, the medium in the flask was replaced with fresh medium until the flask reached 90% cell confluency by visual estimation using a microscope. Cell confluency refers to the degree to which a culture of adherent cells in a flask fills a given area, typically measured as a percentage of the surface area of the flask covered by the cells.

When the cells reach 90% confluency, they are ready for cell passaging. Cell passaging, also known as subculturing, refers to the process of transferring a fraction of cells from an existing cell culture to one or more culture vessels with fresh growth medium to prevent over-confluence, maintain optimal growth conditions, and prolong the culture’s viability and health for extended periods of time [32]. At first, the T75 flask containing cells with the medium was rinsed twice with 10 mL of 1X PBS w/o Ca^2+^, Mg^2+^ solution (Cat. No. BSS-1006-B). The solution was aspirated after each rinse. In the next step, 10 mL of Trypsin-EDTA solution (Sigma T3924) was added to the T75 flask. The flask was swirled to ensure that the Trypsin-EDTA completely covered the surface of the flask. The flask was incubated at 37 °C in a humidified incubator with 5% CO_2_ for 7–8 min. Afterwards, the flask was taken out, and for the next 3 min, the sides of the flask were tapped firmly by fingers to dislodge the cells from the surface of the flask. In total, 15 mL of α-MEM medium was added to the flask to inactivate Trypsin and collect the residual cells. The dissociated cells were transferred to a 50 mL conical tube that was then centrifuged at 800–1000 rpm for 3–5 min. After centrifugation, the supernatant was decanted, and 4 mL of α-MEM medium was slowly resuspended into the cell pellet. After resuspension, the cell concentration in the α-MEM medium was measured using the Auto T4 cell counter (Nexcelom Bioscience, Lawrence, MA, USA) according to the instructions from the cell counter manufacturer. The HBE cell concentration was approximately 4.65 × 10^5^ cells per milliliter of the medium.

### 2.2. Preparation of Bioink

#### 2.2.1. Preparation of Sodium Alginate Solution

The 10% (*w*/*v*) sodium alginate solution was prepared following the procedure described in an early paper [24]. In brief, a 500 mL beaker with 100 mL of deionized water was placed on a hot plate magnetic stirrer (Thermo Fisher, Waltham, MA, USA) that was set to rotate at 800 rpm. Then, 10 g of alginic acid sodium salt powder (Product number: A1112, Sigma-Aldrich, Saint Louis, MO, USA) was slowly added to the beaker for over 2 min, and the solution was stirred for five minutes manually by hand with a stir rod to avoid clumping. Then, the beaker was stirred on a hot plate magnetic stirrer at 60 °C for 2 h. After resting overnight, the solution was stirred again manually by hand for 5 min with a stir rod. The sodium alginate solution in the beaker was sterilized via autoclave at 121 °C for 20 min and was then stored at 4 °C.

#### 2.2.2. Collagen Neutralization

The collagen stock solution (TeloCol-10, type 1 bovine collagen, Advanced Biomatrix, Carlsbad, CA, USA) had a concentration of 10 mg/mL. To prepare 5 mL of neutralized collagen solution at pH 7.4, 4 mL of the collagen stock solution (eight-tenths of the final volume of the resultant collagen solution) was added to a conical tube. Then, 0.5 mL of 10X PBS (one-tenth of the final volume of the resultant collagen solution) was added to the conical tube. After adding 50 µL of 0.5 M NaOH, the pH level of the resultant collagen solution was checked using a pH strip. The volume of 0.5 M NaOH is a variable that depends on the volume of collagen. An additional 37 µL of 0.5 M NaOH was added until the desired pH of 7.4 was achieved. Finally, the total volume of the resultant collagen solution was adjusted by adding 413 µL of deionized water.

#### 2.2.3. Bioink Preparation

Bioink was prepared with the 10% alginate solution (prepared by following the procedure described in Section 2.2.1) and the neutralized collagen solution (prepared by following the procedure described in Section 2.2.2). These two solutions were combined at different ratios to prepare three bioink compositions. Bioink was prepared in batches (2 mL each) for each bioink composition. The first bioink composition was a pure 10% alginate solution. To prepare the second bioink composition, 20% collagen and 80% alginate (or collagen–alginate 1:4 bioink), 0.4 mL of the neutralized collagen solution was added to 1.6 mL of the 10% alginate solution. Finally, to prepare the third bioink composition, 80% collagen and 20% alginate (or collagen–alginate 4:1 bioink) in 1.6 mL of neutralized collagen solution was added to 0.4 mL of the 10% alginate solution.

### 2.3. Preparation of Cell-Laden Hydrogel

Figure 2 illustrates the preparation steps of the cell-laden hydrogel. Three individual 12-well plates were used for three bioink compositions. For each 12-well plate, 3 wells were used for each day’s cell viability assessment for a specific bioink composition. Bioink in an Eppendorf tube was stored in an ice bath prior to the addition of cells to prevent premature thermal gelation. To prepare the cell-laden bioink, 1 mL of the HBE cells with a medium (prepared by following the procedure described in Section 2.1) was added to the bioink to create a final volume of 3 mL, with an effective cell density of approximately 1.55 × 10^5^ cells per milliliter of bioink. Then, using a micropipette (an instrument used to deposit precise volumes of liquid), 250 µL of cell-laden bioink for each composition was added into individual wells of the 12-well plate. Afterwards, the same amount of 300 mM calcium chloride solution was added into the wells with the cell-laden bioink for the ionic crosslinking of bioink, as alginate forms hydrogel in the presence of divalent cations. The addition was performed in a biosafety cabinet. Then, the well plate was incubated at 37 °C in a humidified incubator with 5% CO_2_ for 30 min for the thermal gelation of the bioink, as collagen forms hydrogel at 37 °C. After completing the gelation process, the calcium chloride solution was washed away from the wells, and 1 mL of α-MEM medium was added into the wells with the cell-laden hydrogel. Finally, the well plate was incubated from Day 0 to Day 6 for cell proliferation.

### 2.4. Cell Viability Assessment

In reported studies on the biocompatibility of bioinks for 3D bioprinting [33,34], the live/dead assay method is commonly used to evaluate cell viability [35]. This method involves staining cells with a mixture of fluorophores, where one fluorophore marks live cells and the other marks dead cells, allowing for the visualization and calculation of the percentage of live cells [36].

In this study, an Echo revolution fluorescence microscope (model: RON-K, BICO company) was used to conduct the z-stacking of cell-laden 3D samples to assess cell viability in the samples of three different bioink compositions. Z-stacking, a digital image-processing method, combines multiple images taken at different focal distances to provide a composite image with a greater depth of field [37,38]. The cells were stained with the ReadyProbes™ Cell Viability imaging kit (NucBlue^®^, NucGreen^®^) (Invitrogen™, R37609, ThermoFisher Scientific, Waltham, MA, USA) for fluorescence microscopy. The NucBlue^®^ live reagent stains the nuclei of all live cells and turns the nuclei of the cells highly fluorescent blue in color, which can be detected with a standard DAPI filter of the microscope. The NucGreen^®^ dead reagent stains only the nuclei of dead cells with compromised plasma membranes and turns the nuclei of the dead cells into highly fluorescent green in color, which can be detected with the standard FITC filter of the microscope. The assessment process of cell viability is described below and illustrated in Figure 3.

The well plate was taken out from the incubator and placed in a biosafety cabinet. At room temperature, in the biosafety cabinet, 2 drops of the NucBlue^®^ live reagent and 2 drops of the NucGreen^®^ dead reagent were added to the cell-laden hydrogel in 3 wells of the well plate for a specific bioink composition. The well plate was placed back into the incubator at 37 °C for 15 min for staining to take place. Then, the well plate was taken out from the incubator and placed under a microscope to capture images. For each type of bioink composition, one z-stack was captured using the microscope at randomly chosen positions in the cell-laden hydrogel of each well. The z-stack image files were stored in a computer as ome.tif format. Afterwards, the well plate was kept in the biosafety cabinet, and the live/dead reagents were washed away from the wells. Then, 1 mL of fresh α-MEM medium was added to the wells. Then, the well plate was placed back into the incubator at 37 °C for further proliferation. These steps were repeated over a 6-day period for cell viability assessment on Day 0, Day 1, Day 3, and Day 6.

The ome.tif files of the z-stack images were analyzed using ImageJ Fiji software (version 1.54f). From each z-stack image, 3 planes of focus were randomly selected. The number of live cells (*l_p_*) and the number of dead cells (*d_p_*) in each of the three planes (*p* = 1 to 3) of a particular z-stack image (*z*) were counted. *n_z_* represents the average live cell count of the three planes for each z-stack and was calculated using Equation (1). *N_d_* is the average live cell count of these three z-stacks (*z* = 1 to 3) for a specific bioink composition on a particular day and was calculated using Equation (2).
(1)$$n_{z}=\frac{\sum_{p=1}^{3}l_{p}}{3}$$
(2)$$N_{d}=\frac{\sum_{z=1}^{3}n_{z}}{3}$$

*v_z_* is the average cell viability (%) of the three planes for each z-stack and was calculated using Equation (3). *V_d_* is the average cell viability of the three z-stacks for a specific bioink composition on a particular day and was calculated using Equation (4).
(3)$$v_{z}=\frac{\sum_{p=1}^{3}\left(\frac{l_{p}}{\left({l_{p}+d}_{p}\right)}\times 100\right)}{3}$$
(4)$$V_{d}=\frac{\sum_{z=1}^{3}v_{z}}{3}$$

The average live cell counts of the three z-stacks, *N_d_*_,_ and the average cell viability (%) of the three z-stacks, *V_d_*, were used to assess the compatibility of HEB cells with a specific bioink composition on a particular day.

### 2.5. Statistical Analysis

Statistical analysis was conducted using OriginPro software (version 2024b). Initially, Shapiro–Wilk’s test was used to assess the normality of the data distribution. For the live cell count and cell viability data that passed that normality test, a one-way ANOVA was performed. For the live cell count and cell viability data that failed the normality test, a nonparametric Kruskal–Wallis ANOVA test was performed. Furthermore, Tukey’s post hoc comparison test was used to analyze the differences in experimental data between three types of bioink compositions on Day 0, Day 1, Day 3, and Day 6.

## 3. Results and Discussion

Experimental data are presented in Table 2 and Table 3. To evaluate the statistical significance of the effects of bioink composition for the live cell count on Day 0, Day 1, and Day 6, and cell viability through Day 0 to Day 6, a one-way ANOVA was performed on the experimental data, as these data passed the normality test. The nonparametric Kruskal–Wallis ANOVA test was utilized to determine the statistical significance of the effects of the bioink composition on the live cell count on Day 3, as these data failed the normality test. Table 4 presents the *p*-values for the effects of bioink composition on live cell count and cell viability. Table 5 and Table 6 present the *p*-values from Tukey’s post hoc comparison test for the effects of bioink composition on live cell count and cell viability, respectively.

The effects of bioink composition on live cell count and cell viability are shown in Figure 4 and Figure 5, respectively. The error bars in these figures represent the standard deviation among the three samples for each bioink composition. Figure 4 shows that, on Day 0, Day 1, and Day 6, live cell count was the highest for the collagen–alginate 4:1 bioink composition and the lowest for the alginate bioink composition. Figure 5 shows that the collagen–alginate 4:1 bioink composition provides higher cell viability from Day 0 to Day 6 than the collagen–alginate 1:4 bioink composition and alginate bioink composition. For the alginate bioink composition, cell viability was 71.4% on Day 0 but reduced to 45.0% on Day 6. According to ISO 10993, a reduction in cell viability by more than 30% is considered a cytotoxic effect [34]. For the collagen–alginate 1:4 bioink composition, cell viability was 81.2% on Day 0 and 75.0% on Day 6. The trends observed from this study are consistent with reported trends in the literature. For example, it was reported that a higher alginate proportion in alginate–collagen bioink caused a decrease in cell viability; a higher collagen proportion in alginate–collagen bioink tended to increase the cell viability of human pulmonary lung fibroblasts cells in printed samples [12].

Figure 6 shows that the number of live cells within the cell-laden hydrogel of the collagen–alginate 4:1 bioink composition was noticeably higher than the other two bioink compositions. Clusters of cells could be observed in the images on Day 1, and the clusters gradually became larger from Day 1 to Day 6. In other words, cells proliferated better within the cell-laden hydrogel of the collagen–alginate 4:1 bioink composition than the collagen–alginate 1:4 bioink composition and alginate bioink composition. From the FITC images (with dead cells detected) in Figure 6c, it can be observed that the number of dead cells was higher within the cell-laden hydrogel of the alginate bioink composition compared with the other two bioink compositions. This result indicates that the pure alginate bioink composition is less compatible than collagen–alginate bioink compositions. This means that human HBE cells are more compatible with bioink compositions that have a higher collagen proportion. This is likely because collagen–alginate bioink compositions with a higher collagen proportion contain a greater number of cell-binding ligands, facilitating better cell attachment and interaction [12,39,40]. The higher collagen content may also better mimic the ECM of human bronchial epithelial (HBE) cells, improving the porosity and permeability of the hydrogel and allowing the better diffusion of nutrients and oxygen to the HBE cells.

This study used 16HBE14o-human bronchial epithelial (HBE) cells due to their relevance in respiratory research. HBE cells retain characteristic features of normal differentiated bronchial epithelial cells and serve as a model cell line for airway physiology and diseases like asthma, chronic obstructive pulmonary disease (COPD), and cystic fibrosis. They are crucial for studying mucociliary clearance and barrier integrity. Additionally, HBE cells allow for the investigation of environmental impacts, such as pollutants and pathogens, on the respiratory system. Despite significant advancements in understanding lung function and tissue complexity, how different lung cells respond to extreme environments remains unclear. This understanding is particularly important given the impact of extreme environments, such as those experienced by aircrews, on respiratory health. Aircrews frequently face extreme environments in the cockpit, including pressure changes, vibration, temperature fluctuations, and oxygen deprivation. An aircrew’s responses and performance in extreme environments are evaluated through advanced training, testing, simulations, and real-time monitoring using devices [41,42]. Studies often rely on animal models or monolayer cell cultures, limiting direct applicability to human responses [43,44]. Bioprinting can overcome these issues by embedding cells in a 3D hydrogel, mimicking human tissues more accurately. Additionally, bioprinting improves reproducibility and reduces the need for extensive animal testing approvals.

Before conducting bioprinting, it is essential to find a bioink composition compatible with HBE cells. This study aimed to develop bioinks that are compatible with HBE cells for bioprinting applications. As a key step to selecting a suitable bioink composition, this paper utilized the 3D culture well method, which offers significant advantages over traditional monolayer cultures by more accurately mimicking the in vivo environment. Traditional monolayer cultures typically have adherent cell types that grow submerged in media against a very rigid plastic or glass-culturing surface, where cells are exposed to mostly the same level of nutrients and incubator gas mixture. In contrast, 3D cultures allow cells to grow in a relatively unrestricted environment, with an unequal distribution of oxygen and metabolites. This method enhances cell-to-cell and cell-to-extra cellular matrix (ECM) interactions that are essential for proper cellular function and behavior. Moreover, 3D cultures better replicate signal transduction characteristics and the development of cell polarity in aggregates [45,46,47].

The results of this study are crucial for future research to study how HBE cells respond to extreme environments. By identifying the collagen–alginate bioink composition that is compatible with HBE cells and promotes cell proliferation, this study provides foundational knowledge that enhances the understanding of bioink–cell interactions. Furthermore, this study allows the study of the effects of bioprinting and bioink on cell viability separately. This study provides baseline cell viability data that can then be used to optimize bioprinting parameters. This supports improved bioprinting applications, reduces reliance on animal testing, and paves the way for more accurate and functional 3D-printed constructs, which are essential for studying cell responses to extreme environments.

## 4. Conclusions

This study addresses a critical gap in the current literature by evaluating the compatibility of human bronchial epithelial (HBE) cells in three collagen–alginate bioink compositions, providing essential insights for respiratory tissue engineering. Experimental results highlight the effects of bioink composition on the cell viability of human HBE cells. The collagen–alginate 4:1 bioink composition demonstrated superior performance in maintaining cell viability and promoting cell proliferation compared with the collagen–alginate 1:4 bioink composition and the alginate bioink composition. These results contribute to the foundational understanding of interactions between the collagen–alginate bioink and human bronchial epithelial cells and pave the way for future respiratory tissue studies. Future research should include the characterization of the synthesized material (hydrogel) and 3D-printed constructs.

## Figures and Tables

**Figure 1 bioengineering-11-00862-f001:**
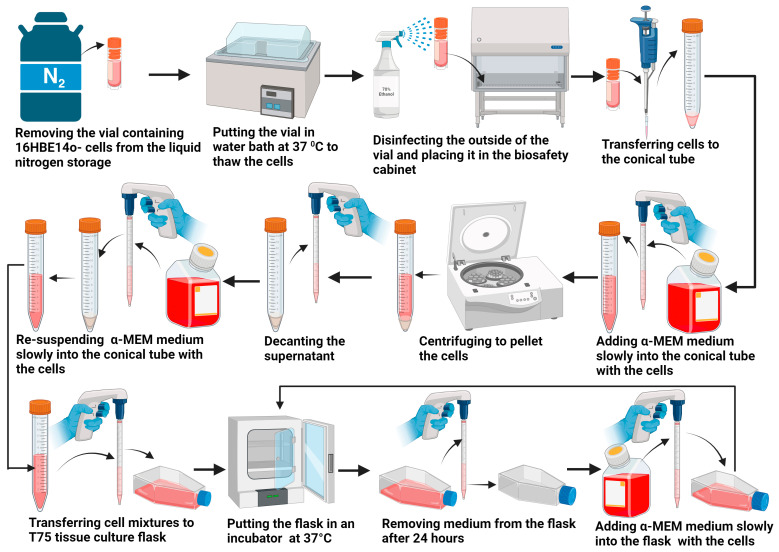
Culturing process for 16HBE14o-human bronchial epithelial cells.

**Figure 2 bioengineering-11-00862-f002:**
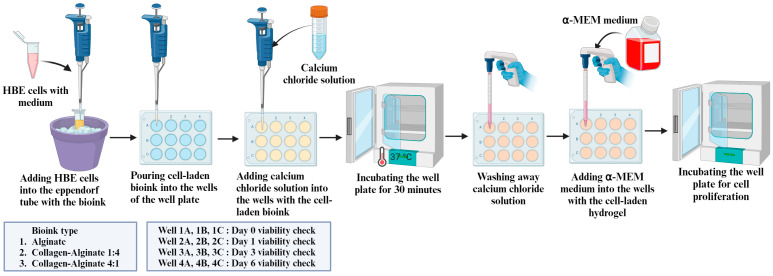
Preparation steps of HBE cell-laden hydrogel.

**Figure 3 bioengineering-11-00862-f003:**
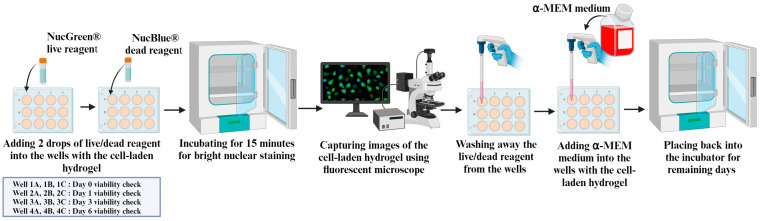
Assessment process of cell viability.

**Figure 4 bioengineering-11-00862-f004:**
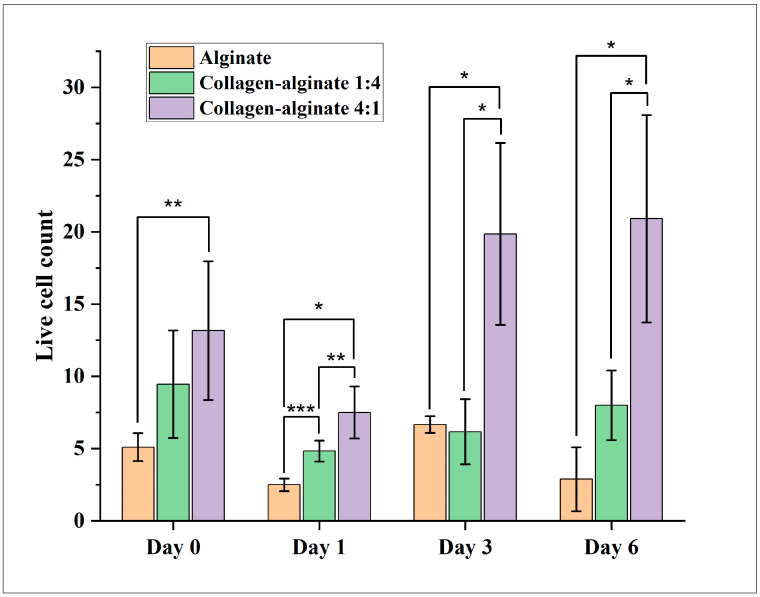
Effects of bioink composition on live cell count (* indicates *p*-value < 0.05, ** indicates 0.05 < *p*-value < 0.085, and *** indicates 0.085 < *p*-value < 0.16).

**Figure 5 bioengineering-11-00862-f005:**
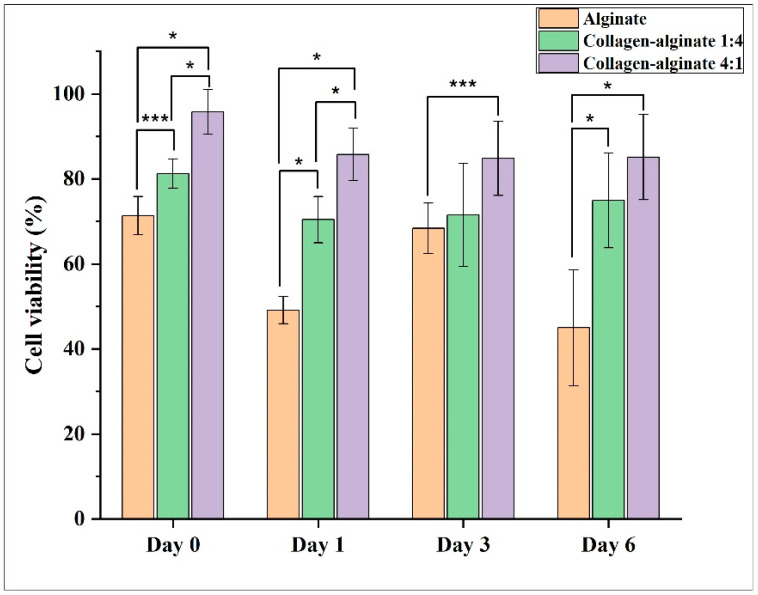
Effects of bioink composition on cell viability (* indicates *p*-value < 0.05, ** indicates 0.05 < *p*-value < 0.085, and *** indicates 0.085 < *p*-value < 0.16).

**Figure 6 bioengineering-11-00862-f006:**
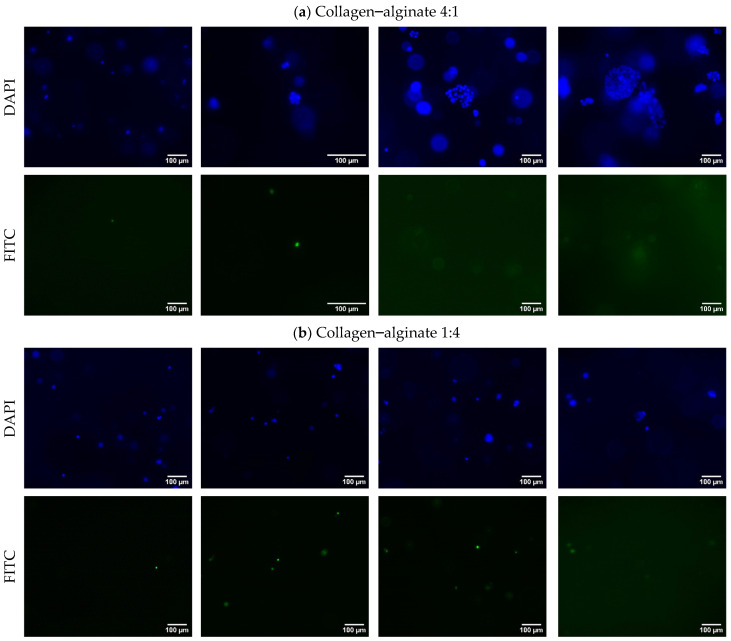
Fluorescence microscopy images of live (blue, DAPI) and dead (green, FITC) cells in the hydrogel of different bioink compositions.

**Table 1 bioengineering-11-00862-t001:** Bioinks used in reported studies on bioprinting with lung cells.

Bioink	Cell Type	Reference
Alginate–collagen	Human pulmonary lung fibroblasts (HPFs)	[12]
Alginate–gelatin	Alveolar types I and II, ciliated, and secretory cells	[2]
Polyvinylpyrrolidone	Human lung epithelial cells(A549), endothelial cells (EA.hy926), and fibroblasts (MRC5)	[17]
Carbopol	Lung cancer epithelial (A549) and lung fibroblast (MRC-5) cells	[18]
Alginate, gelatin, and collagen	Normal human primary lung fibroblasts ((NHLFbs), human monocytic leukemia cells (THP-1), and human epithelial lung carcinoma cells (A549 cells)	[19]
Collagen	Type I and II alveolar cells (NCI-H1703 and NCI-H441), lung fibroblasts (MRC5), and lung microvascular endothelial cells (HULEC-5a)	[20]
Alginate–gelatin	Non-small cell lung cancer PDX cells and lung CAFs	[21]
Alginate–gelatin	Human lung cancer cells A549 and 95-D	[3]
Alginate, gelatin and matrigel	Human alveolar A549 cells	[22]

**Table 2 bioengineering-11-00862-t002:** Experimental data of live cell count for three bioink compositions.

	Alginate		Collagen–Alginate 1:4		Collagen–Alginate 4:1	
	Mean	Standard Deviation	Mean	Standard Deviation	Mean	Standard Deviation
Day 0	5.1	0.96	9.46	3.72	13.17	4.81
Day 1	2.5	0.43	4.83	0.73	7.5	1.80
Day 3	6.67	0.58	6.17	2.26	19.87	6.29
Day 6	2.89	2.22	8	2.40	20.92	7.18

**Table 3 bioengineering-11-00862-t003:** Experimental data of cell viability (%) for three bioink compositions.

	Alginate		Collagen–Alginate 1:4		Collagen–Alginate 4:1	
	Mean	Standard Deviation	Mean	Standard Deviation	Mean	Standard Deviation
Day 0	71.39	4.49	81.23	3.38	95.76	5.23
Day 1	49.13	3.19	70.43	5.45	85.77	6.20
Day 3	68.39	5.90	71.48	12.13	84.83	8.75
Day 6	44.97	13.65	74.96	11.18	85.10	10.01

**Table 4 bioengineering-11-00862-t004:** *p*-values for the effects of bioink composition on live cell count and cell viability.

	Live Cell Count	Cell Viability
Day 0	0.0902	0.0017
Day 1	0.0053	0.0004
Day 3	0.0614	0.1492
Day 6	0.0073	0.0137

**Table 5 bioengineering-11-00862-t005:** *p*-values from Tukey’s post hoc comparison test for the effects of bioink composition on live cell count.

	Day 0	Day 1	Day 3	Day 6
Collagen–alginate 1:4, alginate	0.2731	0.1040	0.9863	0.4100
Collagen–alginate 4:1, alginate	0.0811	0.0043	0.0138	0.0068
Collagen–alginate 4:1; collagen–alginate 1:4	0.6259	0.0663	0.0117	0.0306

**Table 6 bioengineering-11-00862-t006:** *p*-values from Tukey’s post hoc comparison test for the effects of bioink composition on cell viability.

	Day 0	Day 1	Day 3	Day 6
Collagen–alginate 1:4, alginate	0.1066	0.0053	0.9138	0.0460
Collagen–alginate 4:1, alginate	0.0014	0.0003	0.1558	0.0135
Collagen–alginate 4:1; collagen–alginate 1:4	0.0148	0.0241	0.2510	0.5695

## Data Availability

The authors confirm that the data that support the findings of this study are available within the article or upon request to the corresponding author.

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
