# Peer review of "Experimental Study on Compatibility of Human Bronchial Epithelial Cells in Collagen–Alginate Bioink for 3D Printing"

_bioengineering, 2024, doi:10.3390/bioengineering11090862_

Round 1

Reviewer 1 Report

Comments and Suggestions for Authors

In this manuscript the authors assessed the viability of several alginate-collagen formulations as potential bioinks for the culture of human bronchial epithelial cells. The study is short and simple, but its results could significantly contribute to the field. The technical content of this manuscript is good. However, one main weakness may need to be addressed to strengthen its scientific rigor.

Concerns

1.      The authors did not include any statistical analysis.  The p-values when comparing averages must be included within the results and discussion section. In addition, Figures 4 and 5 must include symbols highlighting significant differences between experimental groups.

Reviewer 2 Report

Comments and Suggestions for Authors The manuscript is well written, but I have some minor comments or suggestions to improve the clarity, and scientific soundness of the proposed work."   1.What are the methods used to characterize the synthesized material (hydrogel)?   2. Please add the characteristics of 3D bio printed materials prepared from used alginate the signified the compatibility to used cultured cell?   3. Is there any issue with the stability effect of the enzyme alginate lyase on material stability and cell proliferation?   4. What is the significant utility of this material over the agarose, gelatin, hyaluronic acid, and polyethylene glycol (PEG) diacrylate used in 3D bioprinting?   Also add safety concern/ toxicity concern if any.     

Reviewer 3 Report

Comments and Suggestions for Authors

The present work investigated the compatibility of human bronchial epithelial (HBE) cells in collagen-alginate bioink from a 10% alginate solution and neutralized TeloCol-10 mg/ml collagen stock solution. Cell viability within the hydrogel loaded with HBE cells was assessed using a live/dead assay from Day 0 to Day 6. The experimental results demonstrated that the 4:1 collagen-alginate bioink, highlighting its superior performance in maintaining cell viability and promoting cell proliferation. Although these are very interesting results, the article needs to improve its discussion in the light of the literature, indicating the importance of this study.
Some questions need to be answered:
- Why use this system?
- Why were human bronchial cells (HBE) selected? What is the importance of these results?
 The authors need to critically improve the text if it is to be accepted for publication.

Comments on the Quality of English Language

Nothing to declare.

Round 2

Reviewer 3 Report

Comments and Suggestions for Authors

Dear all,

After reading the new version of the manuscript, I could observe that it improved in quality and scientific aspects. Besides, the authors could answer all questions propely.
In my opinion, the manuscript is ready to publication.